# Critical Role of the Transcription Factor AKNA in T-Cell Activation: An Integrative Bioinformatics Approach

**DOI:** 10.3390/ijms24044212

**Published:** 2023-02-20

**Authors:** Abrahan Ramírez-González, Pedro Ávila-López, Margarita Bahena-Román, Carla O. Contreras-Ochoa, Alfredo Lagunas-Martínez, Elizabeth Langley, Joaquín Manzo-Merino, Vicente Madrid-Marina, Kirvis Torres-Poveda

**Affiliations:** 1Center for Research on Infectious Diseases, Instituto Nacional de Salud Pública, Cuernavaca 62100, Mexico; 2Department of Biochemistry and Molecular Genetics, Feinberg School of Medicine, Northwestern University, Chicago, IL 60611, USA; 3Department of Basic Research, Instituto Nacional de Cancerología, Mexico City 14080, Mexico; 4Consejo Nacional de Ciencia y Tecnología (CONACyT), Instituto Nacional de Cancerología, Mexico City 03940, Mexico; 5Consejo Nacional de Ciencia y Tecnología (CONACyT), Instituto Nacional de Salud Pública, Cuernavaca 03940, Mexico

**Keywords:** AKNA, computational biology, immune response, cell activation, AT-hook transcription factor

## Abstract

The human *akna* gene encodes an AT-hook transcription factor, the expression of which is involved in various cellular processes. The goal of this study was to identify potential AKNA binding sites in genes that participate in T-cell activation and validate selected genes. Here we analyzed ChIP-seq and microarray assays to determine AKNA-binding motifs and the cellular process altered by AKNA in T-cell lymphocytes. In addition, we performed a validation analysis by RT-qPCR to assess AKNA’s role in promoting *IL-2* and *CD80* expression. We found five AT-rich motifs that are potential candidates as AKNA response elements. We identified these AT-rich motifs in promoter regions of more than a thousand genes in activated T-cells, and demonstrated that AKNA induces the expression of genes involved in helper T-cell activation, such as *IL-2*. The genomic enrichment and prediction of AT-rich motif analyses demonstrated that AKNA is a transcription factor that can potentially modulate gene expression by recognizing AT-rich motifs in a plethora of genes that are involved in different molecular pathways and processes. Among the cellular processes activated by AT-rich genes, we found inflammatory pathways potentially regulated by AKNA, suggesting AKNA is acting as a master regulator during T-cell activation.

## 1. Introduction

The AT-Hook motif contains proteins that are involved in diverse biological processes, such as transcriptional regulation, chromosome structure, embryogenesis, and T-lymphocyte differentiation, as well as being involved in pathogenesis of human diseases, including cancer [1,2,3,4]. Additionally, proteins with AT-Hook motifs are involved in regulating the proinflammatory immune response by modulating expression of various cytokines, such as interferon-β (IFN-β), through recognition of AT-rich sites in their promoter regions [5].

AKNA belongs to the AT-Hook transcription factor family [6]. The *akna* gene is localized on the short arm of chromosome 9 (9q32) in a susceptibility locus for cancer and inflammation [6,7]. This gene is 61 kb long and comprises 24 exons [8]. The AKNA protein contains a nine-amino-acid domain (RTRGRPADS), which corresponds to the AT-Hook motif, and also contains a PEST domain (proline, glutamic acid, serine and threonine) for proteolytic degradation. AKNA was discovered in the germinal center of secondary lymphoid organs during the B-cell maturation process and was further described in other cells involved in the immune response, such as T lymphocytes, B lymphocytes, NK cells, and dendritic cells [6]. Thus far, nine isoforms of this gene have been reported (A, B1, B2, C1, C2, D, E, F1, F2), of which only the A and F1 isoforms have been proven to be functional in vitro, showing great capacity for gene expression regulation of T-cell activation costimulatory molecules, such as *CD40* and *CD40 Ligand* (*CD40L*) in B-cells (L612, Nalm6) and T-cells (Jurkat) [4,8]. Therefore, analyzing other genes involved in the mechanisms of T lymphocyte activation, such as *TCR* signaling, in conjunction with costimulatory signals (*CD80*/*CD86*-*CD28*) and the production of cytokines, such as *IL-2*, is a very attractive approach to better understand the role of AKNA in these processes [9,10]. Additionally, AKNA regulates the inflammatory response by inhibiting proinflammatory mediators, such as interleukin-1 β (IL-1 β) and Interferon- γ (IFN- γ), and inflammatory proteins (neutrophilic granule protein, cathelin-related antimicrobial peptide, and S100A8/9) in an in vivo model. Furthermore, when AKNA is silenced, the levels of cytokines and proinflammatory proteins increase [11]. However, an opposite response was observed in an in vitro model (CH3 cells) where proinflammatory cytokines, such as IL-1β, IL-6, IL-11, and TNF-ɑ, showed decreased expression levels following AKNA silencing [12]. This suggests a complex regulatory mechanism dependent on the study model. 

The mechanism of *akna* gene regulation is hypothesized to be determined through pathways regulated by the transcription factors PKA/CREB and NF-κB. A motif prediction analysis of the *akna* promoter performed by Liu X et al. in 2017 demonstrated the presence of binding sites for PKA/CREB and NF-κB/p16 transcription factors in the *akna* promoter region. Additionally, they demonstrated direct binding of PKA/CREB and NF-κB/p16 to the *akna* gene promoter that resulted in inhibited *akna* gene expression. Therefore, the interpretation of the regulatory mechanisms of AKNA in the proinflammatory cytokine pathways and the regulation of the *akna* gene are of great interest for T-lymphocyte activation [12].

T-cell activation is an important process for initiating the immune response against pathogens that attack our organism and in pathologies such as cancer and chronic diseases characterized by alterations in immunity and inflammation. This process involves the expression of various receptors and their respective ligands, diverse signal transduction molecules, and several transcriptions factors. The transcription factors that participate in T-cell activation include AP-1, NF-AT, NF-kB, and SP-1, among others [13]. Moreover, several genes expressed during this process, such as CD40L, IFN-γ, etc., contain an AKNA regulatory element in their regulatory regions [6,9].

Consequently, this information strongly suggests that AKNA may play an important role in the regulation of gene expression during T-cell activation. To this end, we analyzed potential AKNA binding sites in genes that participate in T-cell activation using public data and carried out qPCR validation assays of selected genes to demonstrate AKNA regulation of these genes. We show that AKNA is enriched at AT-rich regions in different genes that are involved in the inflammatory response and that AKNA expression produces increases in expression levels of these genes.

## 2. Results

### 2.1. Prediction of akna Transcription Factor Binding Motifs

We used ChIP-seq data available in the ENCODE database to identify AKNA binding to AT-rich motifs. The ChIP-seq assays were carried out in HepG2 cells that were genetically modified, using the CRISPR-Cas9 editing system, to insert three FLAG tags at the C-terminal end of AKNA (AKNA-flag HepG2), and Flag antibodies were used for immunoprecipitation. The AKNA binding motif analysis was performed in the MEME-ChiP database (Figure 1a). We detected 381 AKNA peaks at promoter regions, as shown in Figure 1b. Enrichment of AKNA-FLAG around transcription start sites (TSS) suggests the presence of AT-rich motifs in promoter regions, making them candidate genes for potential regulation by the AKNA transcription factor.

Based on these data, we carried out an analysis aimed at identifying AKNA-binding motifs in the AKNA-flag enriched regions in the genome of HepG2 cells. For this de novo motif identification analysis, we used the MEME-ChIP tool (Version 5.4.1), using the sequences of the 381 regions enriched for AKNA (enrichment peaks). The MEME-ChIP analysis identified five statistically significant (E value < 0.05) AT-rich motifs as potential AKNA binding sites in the promoter region of various genes to modulate gene expression (Figure 2).

### 2.2. Differential Expression Analysis

Data obtained from the 1-ST Affymetrix Human Gene 1.0 ST Array microarray shows significant alterations in the gene expression profile after activation of T lymphocytes with anti CD3/CD28 and IFN-β/Th-17 cytokines relative to non-activated T lymphocytes (control) at 48 h. The profile of differentially expressed genes is shown in Figure 3. The volcano plot diagram shows a differential gene expression change greater than 1.5 (Fold change) with a *p* value of *p* < 0.05. Approximately 600 genes were found with positive differential expression after T lymphocyte activation, many of which are involved in processes of immune response and cell division. Furthermore, 200 genes were found to be negatively differentially expressed, and these are related to the processes of apoptosis and inhibition of cell proliferation. 

Finally, to demonstrate the association between the presence of AT-rich motifs and differentially expressed genes in T lymphocytes activated with anti-CD3, anti-CD28, and IFN-β/Th-17 cytokines, we selected a few genes involved in proinflammatory processes in activated T lymphocytes and looked at whether AKNA was enriched at their promoter regions. We found that the promoters of *IL-10*, *IL-1α*, *IL-3*, *IFN-γ*, *TNF*, and *IL-2* contain AT-rich motifs. Thus, we believe that the differential gene expression of *IL-10*, *IL-1α*, *IL-3*, *IFN-γ*, *TNF*, and *IL-2* genes will be strongly associated with the recognition of these motifs by the AKNA transcription factor. As an example, in Figure 4, we show the AT-rich motifs in the promoter of *IL-2*. The complete list of AT-rich motifs found in regulated promoter regions can be found in Appendix A. 

### 2.3. Biological Processes and Pathways Affected by T-Lymphocyte Activation

To identify the biological processes and pathways potentially regulated by AKNA, we performed enrichment set analysis of differentially expressed genes using the Enrichr database, considering a *p* value < 0.05 to be statistically significant. The results show heterogeneity in biological processes and signaling pathways (signal transduction, inflammatory response, immune response, and cytokine gene-mediated signaling pathways), as shown in Figure 5.

### 2.4. Expression Analysis of CD80 and IL-2 Genes in Jurkat Cells Transfected with akna

AKNA regulates the expression of genes involved in T-cell activation (*CD40* and *CD40L*) [6]. To further analyze the role of AKNA on the expression levels of genes involved in T-cell activation, we used the Jurkat cell line to quantify the expression levels of *CD80* and *IL-2* genes following *akna* overexpression. Figure 6 shows the average of three independent qPCR analysis experiments, in which we show a statistically significant increase in *IL-2*, and a trend for increased *CD80* gene expression levels in cells transfected with the pcDNA3-*akna* expression plasmid, compared to cells transfected with the empty pcDNA3 vector; *akna*, *CD80*, and *IL-2* gene expression are shown to be 4.1-, 1.19-, and 1.48-fold higher, respectively, in Jurkat cells transfected with *akna*. This suggests that AKNA induces the expression of genes involved in helper T-cell activation, such as *IL-2*, and it is necessary to perform more experiments to strengthen the result that AKNA regulates *CD80* expression, as part of T-cell activation.

## 3. Discussion

In this study we identified five AT-rich motifs in the genome of HepG2 cells, which are potential candidates for recognition by the AKNA transcription factor (ATTT/ACATATATAG/AA, AAATATA, ATAAAAT, AATATTAT, and TATAAA). Additionally, we performed a targeted, intentional search for these AT-rich motifs in the transcriptome of primary T-cells activated with anti-CD3 and anti-CD28 beads as well as IFN-β/Th-17 cytokines and resting cells from healthy individuals. We identified these AT-rich motifs in the promoter regions of genes involved in the processes of cell proliferation, cell cycle, and immune response (*IL-10*, *IL-1α*, *IL-3*, *IL-2*, *INF-γ*, and *TNF*), and genes related with helper T-cell activation, such as *IL-2* and *CD80*. Therefore, as a proof of concept for the validation of genes involved in helper T lymphocyte activation, q-PCR was carried out in the Jurkat cell line, which is a leukemic T-cell line widely used for the understanding of T-cell activation [14]. In the present study, we found that AKNA could positively regulate the expression of a costimulatory molecule, *IL-2*, by direct DNA recognition, and potentially *CD80*. However, further qPCR experiments are needed to demonstrate a significant regulation of AKNA-regulated genes.

CD80 (also known as B7-1) is a member of the B7 family, which consists of structurally related cell surface protein ligands that bind to receptors on lymphocytes to regulate immune response. Activation of T and B lymphocytes is initiated by the involvement of cell surface antigen-specific T-cell receptors or B-cell receptors, but additional signals sent simultaneously by B7 ligands determine the final immune response. These “costimulatory” or “coinhibitory” signals are delivered by B7 ligands through the CD28 family of receptors on lymphocytes. The interaction of CD80 with CD28 performs a costimulatory function that enhances immune responses, while interaction with the cytotoxic T lymphocyte antigen 4 (CTLA4) receptor has a coinhibitory function that attenuates immune responses [15]. Cell surface expression of CD80 and B7-2 (also known as CD86) is down-regulated through transendocytosis by CTLA4. During this process, CTLA4 removes the B7 molecules from the surface of the APCs, thus preventing the interaction of the B7 molecules with the costimulatory molecule CD28 [13,16].

Moreover, IL-2 is a T-cell growth factor required for the regulation of T-cells [17]. IL-2 positively influences homeostasis and development of various T-cell lineages. Expression of interleukin-2 receptor (IL-2R) is restricted to thymically derived regulatory T-cells (nTregs) and antigen-activated T-cells, thus ensuring the specificity of IL-2 activities. IL-2 responsive T-cell subsets have diverse characteristics and pro-inflammatory and anti-inflammatory biological roles. Additionally, IL-2 inhibits Th17 and follicular helper T (Tfh) cell differentiation, making IL-2 an important regulator of T-cell lineage commitment. IL-2 may define the effector/memory fates of CD4+ and CD8+ T-cells; high levels of IL-2 favor the development of short-lived effector cells, while low levels of IL-2 signaling promote differentiation of memory T-cells [18].

As far as we know, AKNA is one of the proteins within the group of molecules with AT-hook motifs that regulates expression of genes involved in lymphocyte activation. In a case and control study, patients with Vogt–Koyanagi–Harada syndrome, characterized by atypical expression of proteins in CD4+ lymphocytes, showed a significant decrease in AKNA and CD18 (integrin B2) proteins, as well as a possible down-regulation of *CD40L* associated with AKNA [4,19].

IL-2 seems to be a potential AKNA-controlled factor whose implications in helper T-cell activation and cancer are relevant for the appropriate function of cells. Nonetheless, there exists evidence indicating that IL-2 is regulated by PI-3K/MAPK or PD-L1 [20,21], and thus these molecules could have a potential impact on AKNA-induced IL-2 effects. Whether AKNA is affected by such regulators remains unknown. A report by Čokić et al., (2015) found that AKNA is down-regulated in myeloproliferative neoplasms where the PI-3K/MAPK pathway is active [22]. Similarly, Liu et al. (2017) reported that MAPK/p38, PKA/CREB, and NF-kB/p65 pathways negatively regulate AKNA expression induced by T-2 toxin in GH3 cells [12].

Conversely, PD-1 and PD-L1 expression correlates with the expression of genes necessary for T-cell activation [23,24]; thus, the regulation of AKNA by PD-L1 is also a possibility. Additionally, the JAK/STAT pathway is involved in the activation of IL-2 in T lymphocytes [25,26]; hence, the possible regulation of AKNA by these pathways cannot be completely discarded, and certainly represents an opportunity for further studies.

In general, genes involved in the primary activation of T lymphocytes initiate a series of signaling cascades that trigger the reorganization of the actin cytoskeleton and centrosome relocation events that are necessary for activation, proliferation, and clonal expansion [27]. With these results and evidence previously cited, we can propose a mechanism of gene expression regulation through the presence of AT-rich motifs in the promoters of genes involved in proinflammatory processes associated with the AKNA transcription factor and its participation in different types of cancer. However, experimental assays are needed to demonstrate the activity of the AKNA transcription factor on potential target genes involved in different molecular pathways and processes, particularly in proinflammatory processes.

Additionally, due to their capacity to bind DNA and modulate gene transcription, proteins with AT-Hook motifs participate in diverse functions, including some human pathologies [1,3]. Some members of the AT-hook transcription factor family, such as ZFAT, PATZ, and AKNA are involved in the immune response. ZFAT is mainly expressed in B and T lymphocytes, the thymus, spleen, and other immune tissues. This protein increases during CD4/CD8 development in the thymus, and regulates genes associated with the immune response [28,29]. Similarly, PATZ is expressed in B lymphocytes, and is involved in T CD4+/CD8+ development [2]. In the same way, AKNA is expressed in several cell types of the immune system, such as T and B lymphocytes, dendritic cells, and natural killers. The role of AKNA in the regulation of costimulatory molecules has been demonstrated during B lymphocyte maturation; in the germinal center, this protein binds to A/T-rich regulatory elements on the promoters of *CD40* and CD40L, regulating their expression, indicating that AKNA participates in the regulation of costimulatory signals in T and B lymphocytes [4,6]. 

The association between the presence of AT-rich motifs in genes implicated in the activation and maturation of cells involved in the immune response (T-cells and B-cells) and the modulation of their gene expression by the transcription factor AKNA through the recognition of AT-rich sites was first shown by Siddiqa et al [6]. Using EMSA assays, they demonstrated that AKNA can positively modulate the expression levels of costimulatory molecules CD40 and CD40L, which are involved in the immunological synapse of T lymphocytes on activation. On the other hand, Ma W et al., in a murine knockout model (C57B/6) with deletion of *akna* exons 19 to 21, observed that knockout mice showed smaller size than wild-type mice and died 10 days after birth due to severe alveolar damage mediated by inflammatory reactions of neutrophils and increased expression of proinflammatory molecules MMP9-1, IL-1β, IFN-γ, CRAMP and S100A9 [11]. This phenomenon of loss of regulation in the expression of these genes could be explained because it has been reported that exon 20 contains the AT-hook site, which binds to the AT-rich motifs in the promoter regions of candidate target genes for negative regulation [8]. In another study, Liu X et al. showed an effect contrary to that reported by Ma W et al., wherein the expression levels of *IL-1β*, *IL-6*, *IL-11*, *TNF-α*, *GH*, and *MMP-9* decreased significantly after AKNA silencing. It is clear that despite the evidence for the possible involvement of the *akna* gene in the regulation of genes involved in molecular processes and mechanisms, the fine and exact mechanism of how AKNA exerts this gene regulatory function remains controversial and at the same time allows the formulation of new hypotheses [11,12]. 

The main strength of this study is that it provides evidence that AKNA could orchestrate the regulation of the expression of a great diversity of genes involved in different molecular and cellular processes. Nonetheless, further studies are required to demonstrate the ability of AKNA to bind to and regulate gene expression to orchestrate an inflammatory response in lymphocytes.

Conversely, one of the limitations of our study is the lack of a model to test the ability of AKNA to bind to AT-rich regions on the indicated genes. This is worth pursuing in future studies.

In conclusion, this integrative bioinformatics analysis suggests that AKNA could modulate the expression levels of genes involved in cell proliferation, immune response, and helper T-cell activation, such as *IL-2* and *CD-80*. Nevertheless, it is necessary to validate the critical role of AKNA in T-cell activation, loss-of-function of AKNA in Jurkat cells, as well as assessment of the immune response in AKNA-deficient Jurkat cells following activation. Those are experiments that we are going to consider to demonstrate the role of AKNA in human T-cell activation, as well as the AKNA-DNA binding experiments after T-cell activation. At present, AKNA silencing tests in Jurkat cells are underway.

## 4. Materials and Methods

### 4.1. Search Strategy

In this study, we searched chromatin immunoprecipitation and massive sequencing (ChIP-seq) data as well as microarray expression data in public repositories such as the Gene Expression Omnibus (GEO) of the National Center for Biotechnology Information (NCBI) “https://www.ncbi.nlm.nih.gov/gds/?term=akna (Accessed on 2 February 2022)” [30] and the Encyclopedia of DNA Elements (ENCODE) of the National Human Genome Research Institute (NHGRI) “https://www.encodeproject.org/ (Accessed on 2 February 2022)” [31].

### 4.2. Database Selection

To determine AKNA-enriched gene regions, the ChIP-seq assay performed on the hepatocellular carcinoma cell line HepG2 was selected from the ENCODE database (accession number ENCSR929IMB) (Accessed on 2 February 2022). The HepG2 cell line was genetically modified using the CRISPR-Cas9 genomic editing system to insert 3 FLAG tags at the C-terminal end of AKNA. With this editing, ChIP-seq assay using a FLAG antibody was achieved.

To associate AKNA binding motifs with gene expression levels, we used microarray expression data corresponding to the 1-ST Affymetrix Human Gene 1.0 ST Array platform obtained from GEO under accession number GSE60235. Expression data correspond to primary T lymphocytes activated with anti-CD3, anti-CD28 beads, IFN-β/Th-17 cytokines, and resting T lymphocytes from healthy individuals.

### 4.3. ChIP-seq Data and Analysis

To determine AKNA gene enrichment in the genome of HepG2 cells, we analyzed chromatin immunoprecipitation and massive sequencing (ChIP-seq) data in duplicate from the human genetically modified (insert) HepG2 cell line expressing AKNA tagged with 3 flag sequences (AKNA-flag HepG2) available from ENCODE. We downloaded the raw data corresponding to AKNA ChIP-seq (accession number ENCSR929IMB) from HepG2 cells. Data analysis was performed on the Galaxy platform [32]. Quality control analysis of raw data was evaluated using FastQC tool “https://www.bioinformatics.babraham.ac.uk/projects/fastqc/ (Accessed on 11 June 2022)” [33] and filtered for quality with the Trim-Galore program “https://www.bioinformatics.babraham.ac.uk/projects/trim_galore/ (Accessed on 11 July 2022)” [34]. Afterward, the reads were aligned and mapped to the reference genome hg19 with Bowtie2, using default parameters [34]. The unmapped and duplicate reads were filtered using the SAMtools tool, with a Q > 20 parameter as the quality score [35]. The AKNA peak enrichment relative to control was determined using MACS2 [36] with the following parameters: region size, 300 bp spread size, and peak detection with a q = 0.05 value. Finally, to visualize the ChIP-seq signal, deepTools2 [37] was used with the parameter binsize 25. To determine the annotation genomic region of the AKNA gene peaks, we used the ChIPseeker package “http://bioconductor.org/packages/devel/bioc/vignettes/ChIPseeker/inst/doc/ChIPseeker (Accessed on 13 July 2022)” [38].

### 4.4. Prediction of AKNA Transcription Factor Binding Motifs

The detection of transcription factor binding motifs at the enrichment regions of AKNA was carried out with MEME-ChiP database [39,40]. A sequence 500 bp upstream and 500 bp downstream of the transcription start site (TSS) was selected. The parameters used were binding motifs between 7 and 25 base pairs of length, from the human and mouse database (HOCOMOCO v11 FULL). MEME ChIP was used to determine motifs with an E-value < 0.05 [39,40].

### 4.5. Microarray Expression Analysis 

To determine whether the gene expression levels are associated with the presence of AT-rich motifs at the promoter regions of candidate target genes of the transcription factor AKNA in primary T-cells activated with anti-CD3, anti-CD28 and IFN-β/Th-17 cytokines, we downloaded the raw data corresponding to these activated primary T-cells available in the 1-ST Affymetrix Human Gene 1.0 ST Array platform at Gene Expression Omnibus GEO (accession number GSE60235).

### 4.6. Functional Enrichment Analysis

To define the biological processes and pathways potentially regulated by AKNA, we used the Enrichr database [41]. We performed an analysis of biological process and pathway enrichment considering a *p*-value < 0.05 as statistically significant.

### 4.7. Validation by qPCR

To validate the expression of possible target genes involved in the immune response, Jurkat cells were transfected with 2.5 µg of plasmids pcDNA3 or pcDNA3-*akna* using PolyFect Transfection Reagent (Qiagen, Hilden, Germany) according to the manufacturer’s instructions. After 24 h of transfection, cells were harvested, and total RNA was purified using RNeasy mini Kit (Qiagen) according to the manufacturer’s instructions. Complementary DNA (cDNA) was synthesized using 1µg of total RNA. Expression levels of *akna*, *CD80*, and *IL-2* genes were performed using 200 ng/µL of cDNA by qPCR technique, using TaqMan inventoried gene expression assay and SYBR Green Master Mix (Applied Biosystems, Foster City, CA, USA), normalizing against the referenced *HPRT1*/*Actin Beta* and according to the manufacturer’s instructions. Data were analyzed using equation 2^−ΔΔCT^ [42]. All primers and the probe were purchased commercially: HPRT1 (QT00083426), AKNA (QT00059066), CD80 (QT00000497), Beta Actin (Hs01060665_g1), and IL-2 (ID Hs00174114_m1).

## Figures and Tables

**Figure 1 ijms-24-04212-f001:**
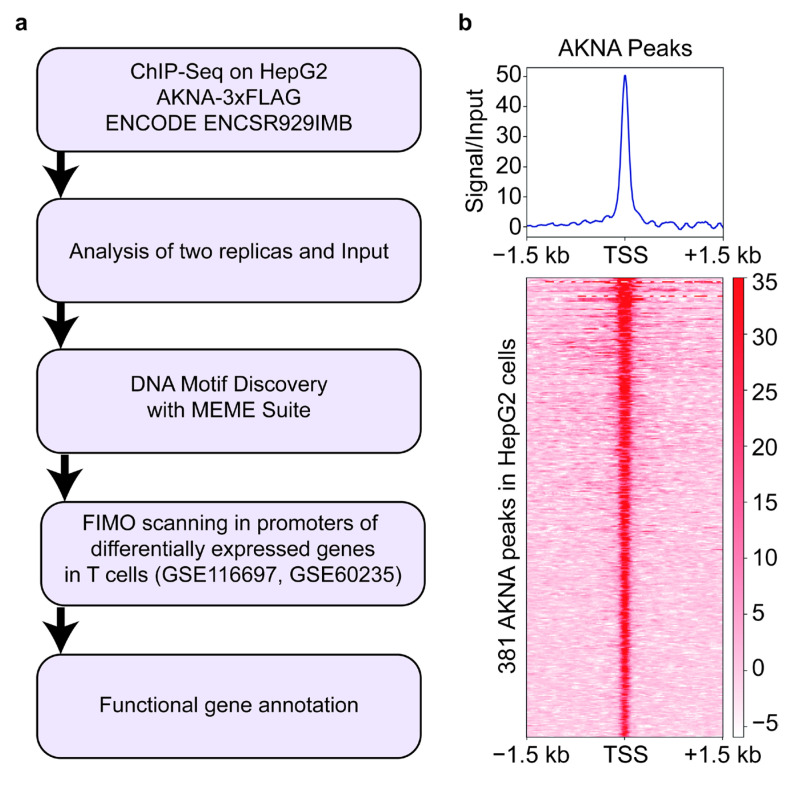
Genomic regions enriched with AT-rich motifs in “AKNA flag HepG2 cells”. (**a**) Flowchart of the methodology for the prediction of AKNA transcription factor binding motifs. (**b**) Heat map of the regions enriched with AT-rich motifs; the signal intensity in red shows the area with the highest enrichment of AT-rich motifs. To validate the presence of homogeneous enriched zones, the graph shows a single peak of AKNA. Peaks detected in the HepG2 cell line are statistically significant (*p* < 0.05).

**Figure 2 ijms-24-04212-f002:**
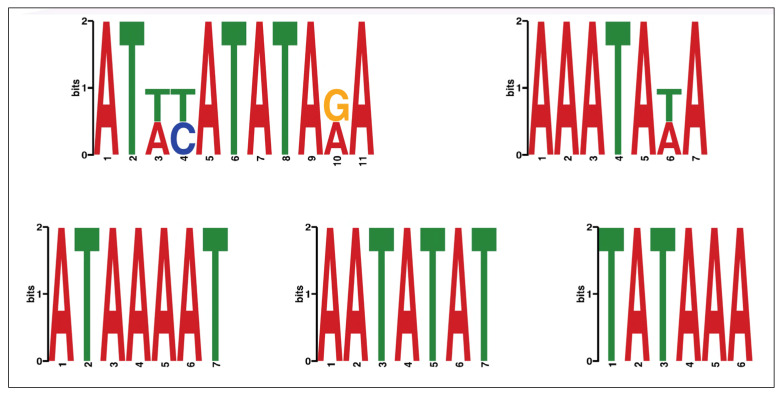
AT-rich binding motifs as potential response elements for the transcription factor AKNA, with possible A/T, T/C, G/A nucleotide changes identified with the MEME-ChIP tool (E value < 0.05).

**Figure 3 ijms-24-04212-f003:**
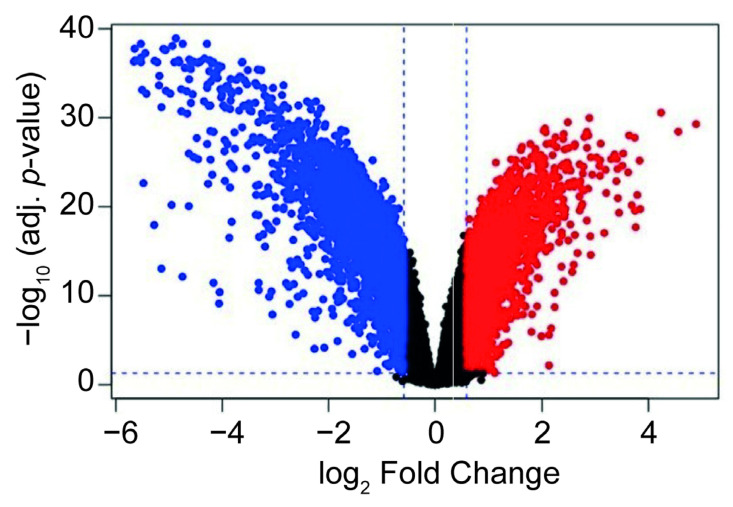
Differential gene expression in activated T lymphocytes. The volcano plot shows the log2 change in the gene expression profile following activation of T lymphocytes with anti-CD3 and anti-CD28 beads and IFN-β/Th-17 cytokines. Each dot represents a gene. Blue dots represent down-regulated genes and red dots indicate up-regulated genes.

**Figure 4 ijms-24-04212-f004:**
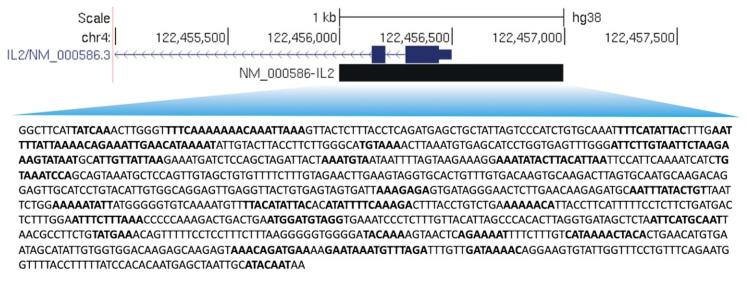
AT-rich motifs in the promoter region of interleukin-2 (*IL-2*). Representative image of AT-rich enriched motifs in the promoter region of *IL-2*, where the promoter region is shown in blue and the AT-rich motifs of *IL-2* are shown in bold.

**Figure 5 ijms-24-04212-f005:**
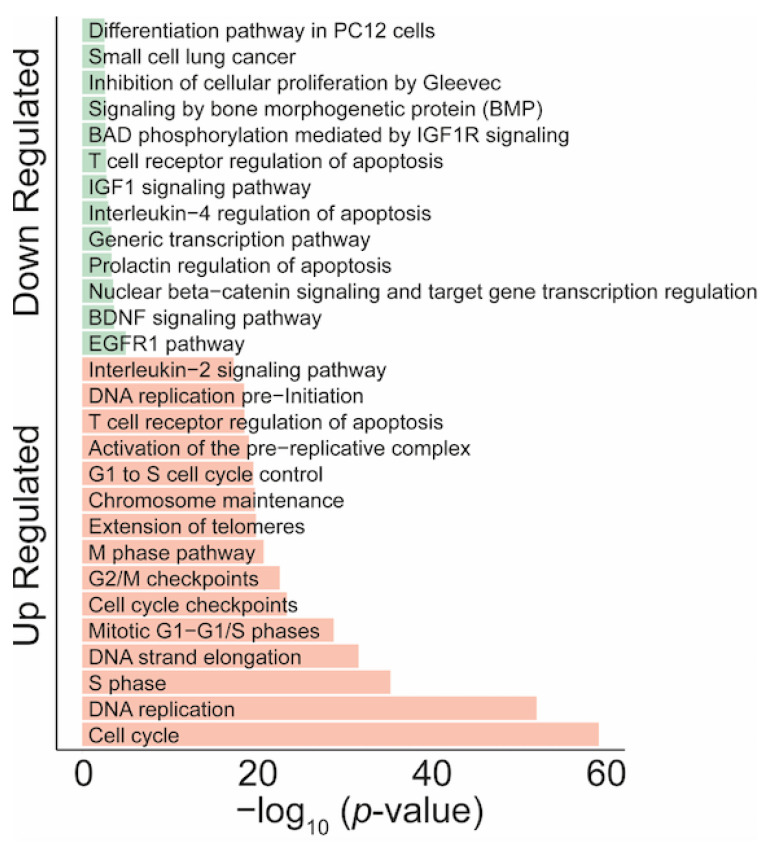
Functional annotation of enriched differentially expressed genes. The most relevant down-regulated pathways (green color) and up-regulated pathways (red color) pathways of genes potentially regulated by the AKNA transcription factor are shown (Enrichr database). An adjusted *p* value < 0.05 is considered to be statistically significant.

**Figure 6 ijms-24-04212-f006:**
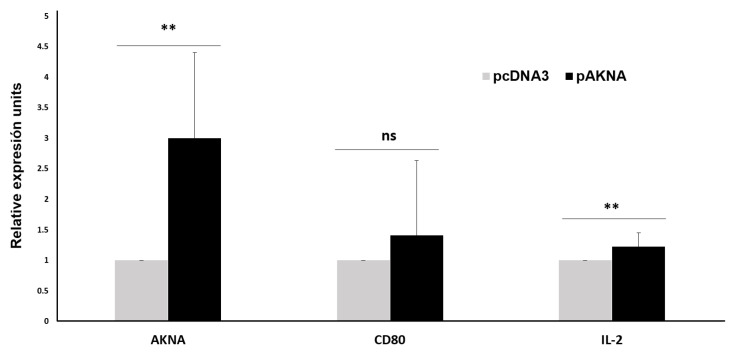
Experimental validation of *CD80* and *IL-2* expression in Jurkat cells transfected with *akna*. *CD80*, *akna* and *IL-2* gene expression levels were measured in Jurkat cells transfected with pcDNA3-*akna* or pcDNA3 plasmids for 24 h. We observed an *IL-2* gene expression compared to the empty plasmid, as well as a trend of increased expression of *CD80* although this did not reach statistical significance. The graph shows the average result from three independent experiments. ns: non-statistically significant, ** *p* < 0.05.

## Data Availability

All databases consulted are indicated in the Materials and Methods section.

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
