# Peer review of "Critical Role of the Transcription Factor AKNA in T-Cell Activation: An Integrative Bioinformatics Approach"

_ijms, 2023, doi:10.3390/ijms24044212_

Round 1
Reviewer 1 Report
Summary:
Abrahan Ramírez-González, et al identified AT-rich motifs that can be potentially recognized by transcription factor AKNA in the genome of HepG2 cells. Additionally, they identified differentially expressed genes in primary T-cells activated with anti-CD3/CD28 beads and IFN-β/Th-17 cytokines relative to non-activated T lymphocytes using publicly available microarray expression data. Finally, they found that differentially expressed genes associated with immune response, such as IL-10, IL-1a, IL-3, TNF, and IL-2 contain AT-rich motifs in their promoter regions. Their data provide implications in the association of AKNA with immune response of T cells upon activation.
General concept comments:
1. To understand the critical role of AKNA in T-cell activation, the dynamic regulation of AKNA in the expression of genes associated with T-cell activation is indispensable. Identification of AKNA binding motif and putative target genes alone does not sufficiently address the knowledge gap. In this manuscript, the authors did not indicate whether there was alteration in the expression level of AKNA and in the binding motif of AKNA upon T-cell activation. Furthermore, to validate the critical role of AKNA in T-cell activation, loss-of-function of AKNA in Jurkat cells as well as assessment of the immune response in AKNA-deficient Jurkat cells following activation need to be performed.
Specific comments:
1. The resolution of the graphs in the manuscript is low.
2. In Figure 1 (a), the indication of genome regions along the X-axis is missing.
Author Response
We edited the manuscript considering the adjustment suggestions of the reviewer 1. The manuscript is attached with control de changes.

Reviewer 2 Report
This is a very interesting work, which can be published in the IJMS upon completion of revisions listed above.
1) In the Introduction, it would be great to see a more extended description of AKNA as a transcription factor and mechanisms of its biochemical regulation.
2) Can the Authors provide details regarding the ChIP sequencing primers used in the studies which they quote?
3) Figure 6 - error bars are not shown, it is thus hard to validate statistical analysis performed. So please revise the diagram.
4) I would suggest the Authors to be more specific about which kind of T cell activation they study. For example, IL-2 is produced by T helpers and is used to activate CD8-positive cytotoxic T cells. So in this (IL-2) case one can only talk about activation of Thelpers (Th1 type of response).
5) In the Discussion, please expand on possible impact of immune checkpoint proteins on AKNA activity. For example, in cancer, recent evidence demonstrated that galectin-9 induces IL-2 expression via PI-3K-dependent pathway, while VISTA blocks this effect (Schlichtner et al. 2023, PMID:36599470). Another review article (Wu et al., PMID: 32152439 summarised the role of PD-L1 in downregulation of PI-3K and MAPK pathways and thus IL-2 production. In light of this, please indicate any exsiting evidence of impact of immune checkpoint proteins on AKNA directly or through their impact on PI-3K and MAPK signalling pathways. And discuss possible (hypothetic) mechanims of involvement of immune checkpoint proteins in regulation of AKNA activity. This is particularly important, since immune checkpoint proteins regulate T cell responses in infection, cancer and autoimmune disease and thus significantly impact signalling events in T cells. As such, AKNA is most likely a part of these networks.
6) I would suggest that the Authors double check the language in the paper or get it read by a native English speaker to improve some stylistic and grammar issues. For example starting from the first sentence in the abstract "The human akna gene encodes an AT-hook transcription factor whose expression participates in various cellular processes" one should better say ".... transcription factor, expression of which is involved in various...."
Author Response
We edited the manuscript considering the adjustment suggestions of the reviewer 2. The manuscript is attached with control of changes.

Round 2
Reviewer 1 Report
the manuscript has been sufficiently improved to warrant publication in IJMS.
Author Response
Response to Reviewers Comments
Date of this review
10 Feb 2023
Note: The comments of the reviewer are in simple text, while our responses are in italic.
The manuscript has improved after revision but there are still some issues that need to be addressed:
Point 1: The address of all the databases used (ENCODE, MEME-ChIP, etc.) need to be inserted in the reference list.
Response 1: Thank you very much for the comment. The adjustment suggested by the reviewer was made. We included all references of the databased used in the references list.
Point 2: Figure 3 appears twice, please remove one.
Response 2: Thank you for your comments. Thank you very much for the comment. The adjustment suggested by the reviewer was made.
Point 3: In the figure legend of figure 4 it is written “ ….. AT-rich motifs of interleukin 2 are shown in yellow …” but in the figure nothing is shown in yellow. May be in bold? Please correct.
Response 3: Thank you very much for the comment. The adjustment suggested by the reviewer in the figure 4 legend was made.
Point 4:
- The results reported in Chapter 2.4 “Expression analysis of CD80 and IL-2 genes in Jurkat cells transfected with AKNA” are very weak.
- Authors start by writing “AKNA regulates the expression of genes involved in T-cell activation (CD40 and CD40L) but then they analyse the expression of CD80 and IL-2. Why?
- Then they continue “Figure 6 shows a representative figure of 3 independent qPCR analysis experiments, in which we clearly demonstrate an increase in IL-2 and a consistent trend for increased CD80 gene …” What does representative mean? If it is a representative experiment I’d imagine that the bars represent the SD of the replicates and it is very strange that there is all this variability if they are just technical replicates. I rather believe that it is the average of three independent experiments. In this case this huge variability would be justified but at the same time it does not allow to conclude that there is a trend of increased expression of CD80, as reported in the legend of figure 4“ …..We observed an increase in CD80 …” This cannot be stated since it is not statistically significant. Also, in the discussion it is reported “….we found that AKNA positively regulates the expression of costimulatory molecules, CD80 and IL-2 …..”
- From Materials and Methods, it appears the gene expression was not normalized for transfection efficiency, this might explain the variability.
- The authors need to perform additional qPCR experiments to demonstrate a significant regulation of AKNA-regulated genes.
Response 4: Thank you for your comments.
- We apologize for stating a result that may need further exploration. We have re-phrased this part in the manuscript to propose a potential regulation without stating an AKNA direct role in gene expression.
- We analyzed CD80 and IL-2 because these are molecules that are expressed upon T-cell activation, and play important role in this process. We focused in testing IL-2 due to the high number of AT-rich motifs found and in fact IL-2 expression it is important to have a complete T-cell activation; if IL-2 expression is weak, there is an incomplete T-cell activation, the T-cell are anergic (Chaplin DD. Overview of the immune response. J Allergy Clin Immunol 2010, pp. S3-S23, 10.1016/j.jaci.2009.12.980). In addition, we already demonstrated that CD40 is regulated by the presence of AKNA (Siddiqa et al 2001).
- We agree with the reviewer, there is not a representative experiment, sorry for the confusion, this figure represents three different experiments, that’s why there is high variability. We agree that CD80 did not reach statistical significance, we need to perform more experiment to make solid the sentence that AKNA regulate CD80 expression, as part of T-cell activation. Still, we demonstrated that IL-2 in deed increases in the presence of AKNA. Several explanations could apply to these results since gene expression could be mediated in a timely-manner producing a sub-observation of the results. We are planning to perform future analysis with this in mind to address the complete history of the expression regulation by AKNA. Finally, the adjustment suggested by the reviewer was made in the figure 6 legend and the lines 256 and 259 of the discussion.
- We agree that we have an important variability, nonetheless, we used HPRT1 and actin as housekeeping genes for normalizing our results. This inconvenient could be addressed extending the number of replicates in the future. This is something we are currently planning to do in the near future.
- We agree with the reviewer, further qPCR experiments are needed to demonstrate a significant regulation of AKNA-regulated genes. This sentence was added in the discussion.
Point 5: line 354: authors write “Additionally, this paper has proposed AKNA as the only protein within the group of molecules with AT-hook motifs that regulates gene expression in lymphocytes”. On which bases was this assumption formulated? HMGA1, another protein containing AT-hook motifs, also regulates gene expression in lymphocytes (56 references in PUBMED). Please remove this statement.
Response 5: Thank you very much for the comment. The adjustment suggested by the reviewer was made.
Point 6: Authors should add in the discussion that, as answered to the first referee, "...loss-of-function of AKNA in Jurkat cells are ongoing to validate the critical role of AKNA in T-cell activation ... "
Response 5: Thank you very much for the comment. The adjustment suggested by the reviewer was made. This sentence was added in the lines 394 to 401 of the conclusion.
We edited the manuscript considering the adjustment suggestions of the reviewer. The last manuscript is attached with control of changes.
